# If You Want to Be Robust, Be Wary of Initialization

**Sofiane Ennadir**[*]
KTH
Stockholm, Sweden

**Johannes F. Lutzeyer**
LIX, Ecole Polytechnique
IP Paris, France

**Michalis Vazirgiannis**
KTH & Ecole Polytechnique
Stockholm, Sweden

**El Houcine Bergou**
UM6P
Benguerir, Morocco

## Abstract

Graph Neural Networks (GNNs) have demonstrated remarkable performance across a spectrum of graph-related tasks, however concerns persist regarding their vulnerability to adversarial perturbations. While prevailing defense strategies focus primarily on pre-processing techniques and adaptive message-passing schemes, this study delves into an under-explored dimension: the impact of weight initialization and associated hyper-parameters, such as training epochs, on a model's robustness. We introduce a theoretical framework bridging the connection between initialization strategies and a network's resilience to adversarial perturbations. Our analysis reveals a direct relationship between initial weights, number of training epochs and the model's vulnerability, offering new insights into adversarial robustness beyond conventional defense mechanisms. While our primary focus is on GNNs, we extend our theoretical framework, providing a general upper-bound applicable to Deep Neural Networks. Extensive experiments, spanning diverse models and real-world datasets subjected to various adversarial attacks, validate our findings. We illustrate that selecting appropriate initialization not only ensures performance on clean datasets but also enhances model robustness against adversarial perturbations, with observed gaps of up to 50% compared to alternative initialization approaches.

## 1 Introduction

Neural networks have demonstrated remarkable prowess across various domains, ranging from computer vision [8] to natural language processing [29], proving their ability to model and extract complex insights from real-world datasets. Recently, Graph Neural Networks (GNNs) [21, 36, 30] have emerged as a powerful extension of neural networks specifically tailored to tackle graph-structured data. These models have led to rapid progress in solving tasks such as node and graph classification where their application have spanned from drug design [20], protein resistance analysis [24], session-based recommendations [33] to tabular data [2]. Concurrently with their success, deep learning architectures have been shown to be unstable when subject to adversarial perturbations [15], resulting in unreliable predictions, consequently questioning these models' applicability in critical domains. While most adversarial robustness studies focus on the domain of computer vision, recent work [16] studying the robustness of GNNs has emerged. Given their rich nature, graphs allow different attack schemes, where the attacker can either choose to edit the graph structure (by adding/deleting edges) or edit the node/edge features. In parallel, recent studies have been devoted to studying approaches to defend against these attacks and enhance GNN robustness, such as input pre-processing techniques [32], low-rank approximation [11], edge-pruning [38] or adapting the message-passing schemes [1].

---

[*]Corresponding Author: `ennadir@kth.se`

38th Conference on Neural Information Processing Systems (NeurIPS 2024).

The majority of available defense studies focus on understanding the inner dynamics of GNNs to pinpoint and mitigate adversarial vulnerabilities. While analyzing the message-passing mechanism and implementing input pre-processing techniques remains a viable direction, comprehensive understanding necessitates exploration beyond traditional avenues. In this sense, investigating factors such as weight initialization strategies and the impact of other hyperparameters, notably those associated with optimization mechanisms, can offer new insights and perspectives on achieving GNN global robustness. Hyperparameter choices and tuning play a critical role in striking a balance between learning the underlying signals in the data and preventing overfitting to ensure the model's generalization. Hence, existing studies on initialization mainly evolve around understanding its effect on the model's convergence, stability and performance [34, 23]. In contrast, our work primarily focuses on examining the effect of initialization on a model's underlying adversarial robustness, representing to the best of our knowledge the first exploration of its kind. Our main objective is to provide a theoretical understanding of the link between weight initialization and other dynamics such as the number of training steps and the resulting model's robustness. With this perspective in mind, we start by formalizing robustness in the context of GNNs when subjected to structural and node feature-based adversarial attacks. Subsequently, we derive an upper bound that connects the model's robustness to the weight initialization strategies. Specifically, we illustrate that this bound depends on the initial weight norms and the number of training epochs. Finally, we validate our theoretical findings by demonstrating the effects of employing various initialization strategies on the model's robustness using benchmark adversarial attacks on real-world datasets. Note that while our analysis primarily focuses on the widely used Graph Convolutional Networks (GCNs) [21] and Graph Isomorphism Networks (GINs) [36], we highlight the versatility of our approach by providing a general upper bound applicable to any Deep Neural Networks in Section 5. This underlines the potential for extending our analysis to a wide range of architectures, showcasing its broad applicability in understanding and enhancing adversarial robustness in neural networks. We summarize our contributions as follows:

- We provide a theoretical analysis that links weight initialization strategies with adversarial robustness in GNNs. We specifically derive an upper bound connecting a model's robustness to weight initialization and the number of training epochs, demonstrating that the initialization strategy can significantly influence the network's adversarial robustness.

- We validate our theoretical findings by conducting extensive experiments across various models using different benchmark adversarial attacks on real-world datasets. These experiments demonstrate that certain weight initialization strategies can enhance the model's defense against adversarial attacks, without degrading its performance on clean datasets.

- While our primary focus is on GNNs, we extend our analysis to Deep Neural Networks, illustrating the broader applicability of our theoretical analysis and its corresponding insights.

## 2 Related Work

**Graph Adversarial Attacks.** Multiple studies focus on designing adversarial attacks capable of fooling a graph-based classifier [16, 35, 10]. The majority of these methods [42, 37] approach the adversarial aim as an optimization problem and employ different methods to solve it such as meta-learning [41]. Furthermore, Nettack [40] constrained the problem by preserving degree distribution and imposing constraints on feature co-occurrence to generate unnoticeable perturbations. Finally, reinforcement learning was proposed recently as a means to generate graph adversarial attacks [7].

**Graph Adversarial Defenses.** Recent efforts have emerged to defend against the aforementioned adversarial attacks. In particular, methods such as low-rank matrix approximation coupled with graph anomaly detection [22] have been used. For example, GNN-Jaccard [32] proposed to pre-process the graph's adjacency matrix to detect potential manipulation of edges. Other methods such as edge pruning [38] and transfer learning [28] have been leveraged to limit the effect of poisoning attacks. Additionally, adaptations of the message-passing scheme, such as employing orthogonal weights [1] or introducing noise during training [9], have been shown to perform well in terms of defense. Furthermore, there is a growing interest in exploring robustness certificates [42, 4] as a means of ensuring model robustness. For instance, [5] used randomized smoothing to provide a highly scalable model-agnostic certificate for graphs. Additionally, other robustness certificates for GCN-based graph classification under topological perturbations have been proposed [19].

**Weight Initialization.** The impact of weight initialization has been extensively studied both theoretically and empirically where the main line of study consists of understanding the interplay between initialization techniques and the implicit regularization they induce, thereby elucidating their influence on a model's generalization capabilities [34, 23]. For instance, it has been showcased that sampling initial weights from the orthogonal group can speed up convergence [18]. Similarly, alternative initialization approaches such as the Glorot Initialization [13] and Kaiming Initialization [17] have been proposed in efforts to improve the model's performance.

Our work stands apart from existing research on adversarial robustness as it represents, to the best of our knowledge, the first attempt to theoretically investigate the impact of initialization on a model's robustness. Moreover, our approach diverges fundamentally from existing literature on weight initialization as our focus lies in theoretically understanding the effect of initialization on a model's robustness rather than its implications for generalization or convergence.

## 3 Graph Adversarial Robustness

In this section, we start by introducing the notation and some fundamental concepts related to GNNs. We afterwards establish the problem setup together with the set of considered assumptions. We finally lay out a GNN's robustness formalization on which we will build our theoretical analysis.

### 3.1 Preliminaries

Let $G = (V, E)$ be a graph where $V$ ($|V| = n$) is its set of vertices and $E$ its set of edges. We denote $A \in \mathcal{A} \triangleq \{0, 1\}^{n \times n}$ its adjacency matrix. The graph nodes are annotated with feature vectors $X \in \mathcal{X} \subseteq \mathbb{R}^{n \times d}$ (the $i$-th row of $X$ corresponds to the feature of node $i$). We denote by $\mathcal{N}(i)$ the neighbors of node $i \in V$ and $\| \cdot \|_2$ the Euclidean (resp., spectral) norm for vectors (resp., matrices).

In this work, we consider the task of node classification. In this task, every node is assigned exactly one class from $\mathcal{C} = \{1, 2, \ldots, C\} \subset \mathcal{Y}$ and we consider $d_{\mathcal{Y}}$ as a distance within the output space $\mathcal{Y}$. The learning objective is to find a function $f_W$, parameterized by $W$, that assigns each node $i \in V$ a class $c \in \mathcal{C}$ while minimizing some classification loss (e. g., cross-entropy loss), denoted as $\mathcal{L}$.

**GNNs.** A GNN model consists of a series of neighborhood aggregation layers that use the graph structure and the node features from the previous layers to generate new node representations. Specifically, GNNs update node feature vectors by aggregating local neighborhood information. In the particular case of GCNs, this process is described by the following iterative propagation:

$$h^{(\ell)} = \phi^{(\ell)} \left( \widehat{A} h^{(\ell-1)} W^{(\ell)} \right), \tag{1}$$

with $W^{(\ell)} \in \mathbb{R}^{p \times q}$ being the weight matrix in the $\ell$-th layer, $p$ and $q$ are embedding dimensions and $\phi^{(\ell)}$ is a non-linear activation function. Moreover, $\widehat{A} \in \mathbb{R}^{n \times n}$ denotes the normalized adjacency matrix $\widehat{A} = D^{-1/2} A D^{-1/2}$, where $D = \mathrm{diag}(|\mathcal{N}(1)|, |\mathcal{N}(2)|, \ldots, |\mathcal{N}(n)|)$ is the degree matrix.

**Problem Setup.** For our theoretical analysis, we assume that the model is based on 1-Lipschitz activation functions (which is a characteristic of commonly used activation functions such as tanh). Additionally, we consider the training loss function $\mathcal{L}$ to be $L$-smooth and that it is minimized using gradient descent. We denote by $W_*$ the local optimum towards which gradient descent iteratively converges. Specifically, for a learning rate $\eta \leq \frac{1}{L}$, the update at time step $t$ for a layer $i$ is:

$$W_{t+1}^{(i)} = W_t^{(i)} - \eta \nabla \mathcal{L} \left( W_t^{(i)} \right).$$

It is worth emphasizing that although we focus on the node classification task, which is prevalent and well-studied in the literature of adversarial robustness, our analysis is equally applicable to other tasks such as graph classification. Moreover, while our theoretical analysis predominantly centers around using gradient descent as the optimizer, this choice does not limit the generality of our findings. One can employ a different optimizer and still yield the same insights and results by following a similar approach as the one outlined in this paper. Consequently, this specific setup should not be perceived as a limitation but rather as an analytical choice.

## 3.2 Adversarial Robustness for Graph Neural Networks

Let $f : (\mathcal{A}, \mathcal{X}) \to \mathcal{Y}$ be a GNN-classifier following the framework outlined in Section 3.1. An adversarial attacks consists of generating an alternative graph $(\tilde{A}, \tilde{X})$ that perturbs the original prediction $f(A, X)$ while not being far (semantically) from the original graph. Typically, this generated graph must adhere to a number of constraints related to its similarity to the original graph, defined by a perturbation budget $\epsilon$ controlling the number of edited edges or features. The set of these graphs is written as $B([A, X]; \epsilon) = \left\{ (\tilde{A}, \tilde{X}) : \min_{P \in \Pi} \left( \|A - P\tilde{A}P^T\|_2 + \|X - P\tilde{X}\|_2 \right) \leq \epsilon \right\}$, where $\Pi$ represents the set of permutations of the adjacency matrix. While the previous formulation relies on the $\ell_2$ norm, other norms may be used depending on the domain of application and the specific use case. Building on previous work [9], the adversarial risk of a GNN can be defined as the expected error of adjacent graphs within the considered graph's neighborhood defined by $\epsilon$ written as:

$$\mathcal{R}_\epsilon[f] = \mathop{\mathbb{E}}_{(A,X) \sim \mathcal{D}} \left[ \sup_{(\tilde{A}, \tilde{X}) \in B([A,X]; \epsilon)} d_\mathcal{Y} \left( f\left(\tilde{A}, \tilde{X}\right), f(A, X) \right) \right]. \tag{2}$$

In the current analysis, we focus on the $\ell_2$ norm as our output distance $d_\mathcal{Y}$ (which can be substituted by any norm – given the equivalence of norms). We theoretically approach the introduced adversarial risk by deriving an upper-bound, which reflects the model's expected error under input perturbation. Intuitively, a smaller upper bound reflects a smaller adversarial risk which in turn suggests a robust behavior locally. In this perspective, Definition 1 draws the link between the considered risk quantity and a model's robustness.

**Definition 1.** (Adversarial Robustness). The graph-based function $f : (\mathcal{A}, \mathcal{X}) \to \mathcal{Y}$ is said to be $(\epsilon, \gamma) -$ robust if its adversarial risk is upper-bounded by $\gamma$, i.e., $\mathcal{R}_\epsilon[f] \leq \gamma$.

The current definition addresses adversarial risk from a worst-case scenario perspective, which is the most prevalent approach in the literature. This means we aim to identify the neighbor graph that maximizes the harm (i.e., causes the greatest deviation from the original prediction). By upper-bounding the risk associated with this "worst-case" graph, we inherently account for all other potential adversaries within the same neighborhood, as their risk will be less than or equal to that of the worst-case scenario. We note that the nuances between the "average" and "worst-case" approaches have been thoroughly examined and justified in previous research [25].

## 4 On the Effect of Initialization

We start by considering the Graph Convolutional Networks (GCNs) within the broader context of Message Passing Neural Networks for node classification. This study investigates how initialization and other hyperparameters impact the final model's robustness. In this context, we aim to establish a connection between the introduced adversarial risk (Equation (2)) and the initial weight distribution and its evolution during training. Specifically, we seek to demonstrate that different choices in the initialization distribution and other relevant parameters lead to varying levels of model robustness, offering new insights into the potential trade-offs between initialization strategies and robustness. In this sense, we derive an upper-bound (denoted as $\gamma$ in Definition 1) on the stability of a GCN-based classifier when the input graph's node features are subject to adversarial attacks.

**Theorem 2.** *Let $f : (\mathcal{A}, \mathcal{X}) \to \mathcal{Y}$ denote a graph-based function composed of $T$ GCN layers, where the initial weight matrix of the $i$-th layer is denoted by $W_0^{(i)}$. For adversarial attacks only targeting node features of the input graph, with a budget $\epsilon$, we have (in respect to Definition 1):*

$$\gamma = \epsilon \prod_{i=1}^{T} \left( 2^t \left\| W_0^{(i)} \right\| + 2^{t+1} \left\| W_*^{(i)} \right\| \right) \left( \sum_{u \in \mathcal{V}} \hat{w}_u \right)$$

*with $t$ being the number of training epochs and $\hat{w}_u$ denoting the sum of normalized walks of length $(T-1)$ starting from node $u$.*

The proof of Theorem 2 is provided in Appendix A. Theorem 2 provides a formal connection between the robustness of a GCN-based classifier and its initial weights, offering valuable insights into their

effects. From a first perspective, the derived upper-bound depends on the initial weight's norm. Specifically, a lower norm corresponds to a smaller upper-bound, indicative of a more robust model. However, while setting all initial weights to zero theoretically yields the smallest upper-bound and consequently the optimum robustness, this direction can detrimentally affect the model's performance on the learning task. Empirical evidence suggests that initializing weights to zero (or a constant) often leads to poor learning outcomes, as it constrains weight behavior during propagation, limiting subsequent back-propagation operations and resulting in convergence to unsatisfactory local minima (e. g., see Page 301 in [14]). From a second perspective, it appears that a higher number of training epochs leads to the looseness of the upper-bound, resulting in increased adversarial vulnerability. This latter observation provides proof and highlights the existence of the usually discussed trade-off between clean and attacked accuracy. Achieving a balance between increasing the number of epochs to achieve satisfactory clean accuracy and limiting them to attain a robust model is hence essential. While theoretically challenging to identify this equilibrium point, our experimental results demonstrate its existence. We note that the dependence of $\gamma$ on $t$ can be sharpened by having $(1 + \eta L)^t$ instead of $2^t$. With small $\eta$ (which is usually the case in practice), $(1 + \eta L)^t \approx 1 + t\eta L$ resulting in a bound which depends linearly on $t$. The same remark applies to the remaining bounds derived in the paper. These insights, in the case of node-feature-based adversarial attacks, also extend to structural perturbations where Theorem 3 provides the exact bound for this case.

**Theorem 3.** *Let $f : (\mathcal{A}, \mathcal{X}) \to \mathcal{Y}$ denote a graph-based function composed of $T$ GCN layers, where the initial weight matrix of the $i$-th layer is denoted by $W_0^{(i)}$. Let $f$ be the number of used training epochs. When $f$ is subject to structural attacks, with a budget $\epsilon$, we have (in respect to Definition 1):*

$$\gamma = \epsilon \prod_{i=1}^{T} \left( 2^t \left\| W_0^{(i)} \right\| + 2^{t+1} \left\| W_*^{(i)} \right\| \right) \|X\| \left( 1 + T \prod_{i=1}^{T} \left( 2^t \left\| W_0^{(i)} \right\| + 2^{t+1} \left\| W_*^{(i)} \right\| \right) \right).$$

The computed upper-bound suggests that the effect of initialization is greater in the case of structural perturbations. This emphasis is resulting from the distinct dynamics within the message passing mechanism, where the influence of the adjacency matrix and node features varies during each propagation step. Precisely, for structural perturbations, the effect of the attack is considered at each propagation step through the perturbed adjacency matrix (in the aggregation step). Moreover, the impact is also amplified by the affected residual layers from previous iterations, resulting in a more significant attack result. This is different in the case of node-feature based adversarial attacks, since the node features are only directly taken into account in the first propagation. Overall, the main takeaway of the provided analysis in Theorems 2 and 3 is that "approximately-free" robustness enhancements can be derived from choosing the right initial weight's distribution and the right number of training epochs. We illustrate this specific point by analyzing the effect of the initial distributions choices on the model's robustness. Specifically, we consider the case of the Gaussian distribution, where Lemma 4 studies how the parameters of this distribution – namely, the mean and variance – exert an influence on the expected (in respect to the initial distribution) value of the adversarial risk.

**Lemma 4.** *Let $f : (\mathcal{A}, \mathcal{X}) \to \mathcal{Y}$ denote a graph-based function composed of $T$ GCN layers for which the initial weight are drawn from the Gaussian distribution $\mathcal{N}(\mu, \Sigma)$. When subject to node features based adversarial attacks, we have the following:*

$$\mathbb{E}_{W_0 \sim \mathcal{N}(\mu, \Sigma)}[\mathcal{R}_\epsilon[f]] \leq \epsilon \prod_{i=1}^{T} \left( 2^t \sqrt{\mu^2 + tr(\Sigma)} + 2^{t+1} \left\| W_*^{(i)} \right\| \right) \left( \sum_{u \in \mathcal{V}} \hat{w}_u \right).$$

The proof of Lemma 4 is provided in Appendix C. Given that a tighter upper bound inherently results in a higher level of robustness, the results derived in Lemma 4 illustrate the clear effect of initialization in the case of the Gaussian distribution. The derived bound shows that increasing the distribution parameters, both the mean and variance values, leads to a decrease in the victim model's underlying robustness. While one might intuitively aim to set these parameters as low as possible to achieve optimal robustness, doing so could potentially compromise the model's performance on clean datasets. Therefore, as previously mentioned, striking the right balance between clean accuracy and adversarial robustness is crucial.

**Extending the Results to the GIN.** The same previously applied analysis for the GCN-based models can be extended to take into account GIN-based classifiers. We consider the same set of assumptions and the same problem setup considered during the previously studied GCN case. We additionally

assume that the input node feature space to be bounded, i. e., $\|X\| \le B$. We note that this boundedness is a realistic assumption and that the value $B$ can be easily computed for any real-world dataset.

**Theorem 5.** *Let $f : (\mathcal{A}, \mathcal{X}) \to \mathcal{Y}$ denote a graph-based function composed of $T$ GIN layers, where the initial weight matrix of the $i$-th layer is denoted by $W_0^{(i)}$. For adversarial attacks only targeting node features of the input graph, with a budget $\epsilon$, we have:*

$$\gamma = \prod_{l=1}^{T} \left( 2^t \left\| W_0^{(i)} \right\| + 2^{t+1} \left\| W_*^{(i)} \right\| \right) \left[ BT \max_{u \in \mathcal{V}} deg(u) + \epsilon \right]$$

*with $t$ being the number of training epochs and $deg(u)$ is the degree of node $u$.*

The proof of Theorem 5 is provided in Appendix D. Theorem 5 establishes an upper bound on the robustness of a GIN-based classifier against adversarial attacks targeting node features. We observe analogous insights, to the ones derived for a GCN-based classifier, regarding the influence of the initialization distribution and number of training epoch on the model's underlying robustness.

# 5 Generalization to Other Models

While our primary research focus lies within the domain of graph representation learning, a sub-field of the broader landscape of Deep Learning models, the fundamental principles of our theoretical analysis are applicable across various model architectures. Notably, and to our knowledge, the absence of a comparable study in current adversarial literature motivates our endeavor to bridge this gap. In this section, we aim to fill this gap by presenting a comprehensive analytical framework that provides the connection between weight initialization and the robustness of neural networks.

Let $x \in \mathbb{R}^{n_0}$ denote an input vector where $n_0$ is the input dimension. Let $W^{(l)} \in \mathbb{R}^{n_{l-1}, n_l}$ be the weight matrix and $b_l \in \mathbb{R}^{n_l}$ the bias of the $l^{\text{th}}$ layer with $n_l$ being its dimensionality. We focus on the general family of neural networks for which the computation during layer $l$, using an activation function $\phi^{(l)}$, can be written as :

$$h^{(l)} = \phi^{(l)} \left( W^{(l)} h^{(l-1)} + b^{(l)} \right) .$$

We consider the same set of assumptions (stated in Section 3.1) as the one from previous section. We consider the $\ell_2$ norm as our input and output distances within the metric space $\mathbb{R}^{n_0}$ and we consider an input attack budget $\epsilon$. The introduced adversarial risk in Equation 2 can be easily extended and tailored to the family of considered neural networks discussed in this section. Further clarification on this extension is provided in the Appendix (Section G.1). From this standpoint, by adapting the Definition 1, analogous effects of the weight initialization, provided in Theorem 6, can be observed.

**Theorem 6.** *Let $f : \mathcal{X} \subseteq \mathbf{R}^{in} \to \mathcal{Y} \subseteq \mathbf{R}^{out}$ be a $T$-layers neural network with $W_0^{(i)}$ denoting the initial weight matrix of the $i$-th layer. When subject to adversarial attacks, $f$ is $(\epsilon, \gamma) - robust$ with:*

$$\gamma = \epsilon \prod_{i=1}^{T} \left( 2^t \left\| W_0^{(i)} \right\| + 2^{t+1} \left\| W_*^{(i)} \right\| \right)$$

The proof of Theorem 6 can be found in Section E of the Appendix. Similar to previous findings, the upper bound relies on key elements of the initialization process, specifically the initial weight norm and the number of training epochs. These results validate and extend the established link between initialization and a model's robustness in neural networks, highlighting the importance of selecting appropriate parameters. From the derived upper bound, which is also applicable to GCN and GIN cases, we observe that the number of training epochs exerts an effect on the bound. Specifically, while increasing the number of epochs can improve the model's performance on a clean dataset, it simultaneously leads to a deterioration in the model's adversarial robustness. Ideally, adversarial defense strategies aim to avoid this trade-off between clean and attacked accuracy, striving for robust models that do not compromise the initial performance. In this context, considering the strong-convexity of the loss function $\mathcal{L}$, in addition to the previously made assumptions, we observe that the effect of the number of training epochs becomes less pronounced. Lemma 7 specifically provides the computed bound under these assumptions.

**Lemma 7.** *Let $f : \mathcal{X} \subseteq \mathbf{R}^{in} \to \mathcal{Y} \subseteq \mathbf{R}^{out}$ be a $T$-layers neural network trained with a $\mu$-strongly convex and $L$-smooth loss function. Let $W_0^{(i)}$ denote the initial weight matrix of the $i$-th layer. When subject to adversarial attacks, with a budget $\epsilon$, we have that $f$ is $(\epsilon, \gamma) - robust$ with:*

$$\gamma = \epsilon \prod_{i=1}^{T} \left( (1 - \mu/L)^t \left\| W_0^{(i)} \right\| + 2 \left\| W_*^{(i)} \right\| \right)$$

The proof of Lemma 7 is provided in Section F of the Appendix. Since $\mu \leq L$, increasing the number of training epochs results in the diminishing influence of the initialization weights. In this scenario, the bound depends solely on the final weights, a phenomenon previously explored in works such as Parseval networks [6] for neural networks and GCORN [1] for GNNs. This observation highlights the necessity of convexity in the loss function when training a neural network, as it plays a crucial role in enhancing the model's robustness, beyond the traditional considerations of classical training optimization perspectives.

## 6 Experimental Results

This section aims to empirically validate our theoretical findings using real-world benchmark datasets. We start by laying out our experimental setting, then we study the impact of various initialization strategies on a GCN's robustness. Next, we analyze the influence of training epochs on adversarial robustness. Finally, we extend our experimentation to considered family of DNNs in Section 5.

### 6.1 Experimental Setting

**Experimental Setup.** Consistent with our theoretical analysis, this section focuses on the node classification task. We leverage the citation networks Cora and CiteSeer [27], with additional results on other datasets provided in the Appendix G. To mitigate the impact of randomness during training, each experiment was repeated 10 times, using the train/validation/test splits provided with the datasets. A 2-layers GCN classifier with identical hyperparameters and activation functions was employed across all the experiments. The models were trained using the cross-entropy loss function, and consistent values for the number of epochs and learning rate were maintained across all analysis. Further implementation details can be found in Appendix H. The necessary code to reproduce all our experiments is available on github https://github.com/Sennadir/Initialization_effect.

**Adversarial Attacks.** We consider two main gradient-based structural adversarial attacks: **(i)** 'Mettack' (with the 'Meta-Self' training strategy) [41] that formulates the problem as a bi-level problem solved using meta-gradients **(ii)** and the Proximal Gradient Descent (PGD) [35] which consists of iteratively adding small crafted perturbations using the gradient of the classifier's loss. We additionally provide results for the 'Dice' attack [41] in Appendix G. For our experiments, we considered perturbation rates ranging from $10\%$ (i. e., $0.1|E|$) to $40\%$ (i. e., $0.4|E|$).

**Evaluation Metrics.** We report the experimental findings in terms of the 'Attacked Accuracy', which is the model's test accuracy when subject to the attacks. Additionally, given that initialization have an impact on the model's generalization and performance, solely reporting the attacked accuracy fails in some specific cases to provide a comprehensive perspective. Thus, we adopt for some experiments the "Success Rate" metric, also commonly employed in adversarial literature, which encompasses the number of successfully attacked nodes while taking into account the model's initial clean accuracy.

### 6.2 Effect Of Training Epochs

The theoretical analysis presented in Section 4 established a connection between the number of training epochs and the model's resulting robustness. The derived bound suggests that increasing the number of epochs results in the model becoming more vulnerable to adversarial attacks. The objective of this experimental section is to empirically validate this assertion using real-world datasets. To this end, at each training epoch, we assess the model's performance on the test set, considering both its clean accuracy and its accuracy under adversarial attacks.

Figure 1 illustrates the results of this analysis. The initial two subplots (a,b) display the findings on the Cora dataset, while the subsequent (c,d) subplots present results from the CiteSeer dataset. For

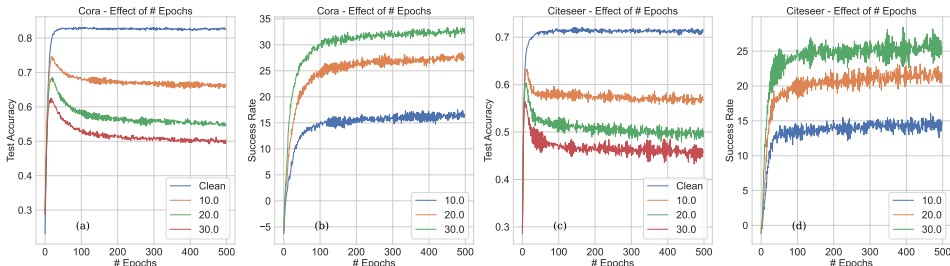

Figure 1: Effect of training epochs on the model's robustness on Cora (a,b) and CiteSeer (c,d).

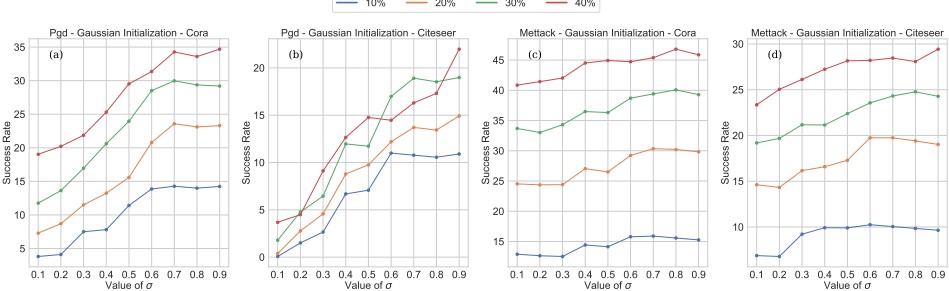

Figure 2: Effect of the variance parameter on the model's robustness in the case of Gaussian Initialization on PGD [on Cora (a) and Citeseer (b)] and Mettack [on Cora (a) and Citeseer (b)].

each dataset, the first plot showcases the clean and attacked accuracy, while the second plot shows the Success Rate (the discrepancy between the clean and attacked accuracy for each budget). The experimental results demonstrate the existence of the previously discussed trade-off between clean and robust accuracies. Specifically, as anticipated, the clean accuracy exhibits a continual increase until reaching a plateau, corresponding to the convergence of the loss function to a minimum. Conversely, the attacked accuracy demonstrates a rising trend until reaching an inflection point, beyond which it begins to decline. These findings confirms the observations from the derived upper-bound, indicating that a higher number of epochs leads to increased vulnerability in the model. Ideally, users would aim to stop training at the inflection point, where the attacked accuracy is maximized while the clean accuracy remains proximal to its convergence point.

## 6.3 Effect Of Initial Weight Distribution

We aim to validate the impact of the initial weight norms on the model's adversarial robustness. As previously discussed in Section 4, a larger weight norm leads to the relaxation of the upper-bound, potentially resulting in the model being more susceptible to adversarial attacks.

In this perspective, we start by investigating the effect of sampling from a Gaussian distribution, as studied in Lemma 4. We hence consider this latter by setting the mean value $\mu$ to a constant, and analyzing the impact of the variance parameter $\sigma$. Intuitively, based on the upper-bound analysis, a higher variance value is anticipated to result in reduced model robustness. Figure 2 illustrates the resulting Success Rate across various variance values for both the "PGD" and "Mettack" methods, applied to the Cora and Citeseer datasets. The findings unequivocally validate the theoretical insights, demonstrating a direct correlation between increasing the variance ($\sigma$) and a higher Success Rates, indicating heightened vulnerability and reduced robustness of the model. Moreover, the impact of initialization becomes more pronounced when considering larger attack budgets, as outlined in the computed upper-bound. Notably, for certain budgets (e.g., $30\%$ and $40\%$), the observed gap ranges between $5\%$ and $15\%$, underscoring the initial weights significant implications on the robustness.

Within the same context, we explore alternative initialization strategies, focusing on two primary cases. First, we investigate sampling initial weights from a uniform distribution $\mathcal{U}(-\beta, \beta)$, where $\beta$ can be seen as a scaling parameter for weight norms. Second, we consider employing a scaled orthogonal weight initialization strategy. While this our aim can be approached by sampling weights

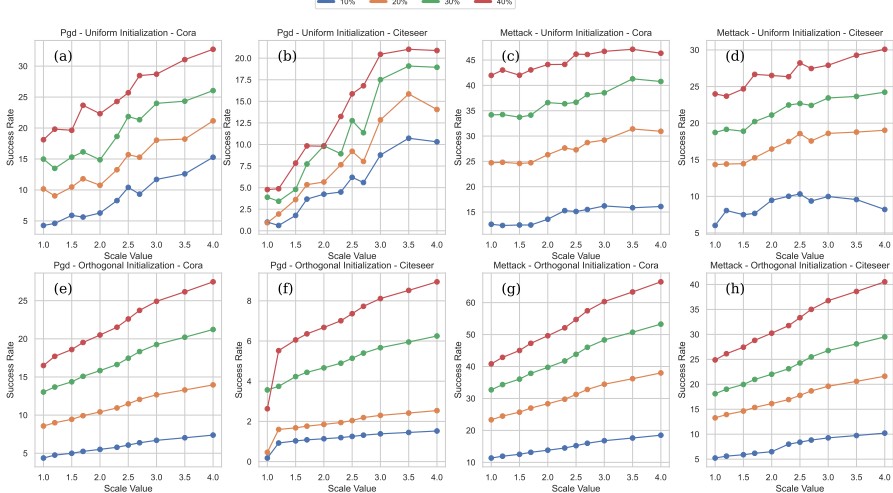

Figure 3: Effect of the scaling parameter $\beta$ on the model's robustness in the case of Uniform (a-d) and Orthogonal (e-h) Initialization when subject to PGD and Mettack using Cora and CiteSeer.

from a scaled random Gaussian distribution, we adopt the orthogonal initialization strategy proposed in prior work [26], which we further rescale by a factor $\beta$ to examine the impact on weight norms. In both cases, higher scaling parameter values of $\beta$ are anticipated to theoretically yield higher upper-bounds and consequently render the model more vulnerable, as indicated by our computed bounds. We conduct numerical computations on both the Cora and Citeseer datasets to assess the resulting adversarial robustness of a GCN across various $\beta$ values, as provided in Figure 3. The experimental results are exactly aligned with our theoretical findings showcasing the effect of the weight norm in the adversarial robustness. To summarize, while traditionally overlooked in prior studies on adversarial robustness, our experimentation underscores the critical importance of selecting appropriate initialization distributions and strategies for enhancing model robustness.

## 6.4 Experimental Generalization

We extend our experimentation to empirically validate the theoretical generalizations provided in both Section 4 for the GINs and Section 5 for a DNNs. To this end, we consider these two models with various initialization schemes, including the previously used Orthogonal [26] and Uniform initialization in addition to the Kaiming [17] and Xavier Initialization [13]. Our analysis primarily focuses on the PGD adversarial attack, using iden-

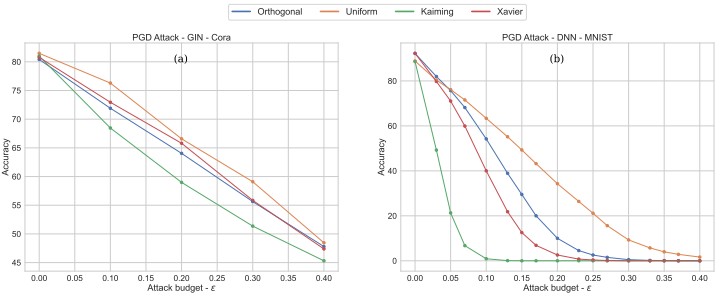

Figure 4: Effect of initialization on the GIN (a) and DNN (b) for different attack budgets.

tical attack budgets as in the previous sections. Figure 4 presents the results on the GIN (a) using the Cora dataset and (b) on the DNN using the MNIST dataset. Notably, we observe that the different initialization methods yield similar clean accuracy ($\epsilon = 0$), yet as the attack budget increases, the discrepancy in attacked accuracy between them also grows. For instance, in the case of DNNs, the accuracy gap between the best and worst initialization methods for $\epsilon = 0.1$ ranges around $60\%$, proving our main assumption related to the impact of initialization on the model's robustness.

# 7    Conclusion & Limitations

The current study shows that the dynamics of learning in GNNs and DNNs have an important effect on the model's final robustness. Specifically, we theoretically showed that the model's robustness is connected to the weight initialization and the number of training epochs. We empirically validate our findings, where we can see that choosing the right initialization can yield huge "almost-free" robustness improvement. We additionally showed the existence of a trade-off between choosing the right number of epochs to have the best clean accuracy and the most robust model. While the current work did not propose an alternative or a solution, it has introduced a new perspective, which to our knowledge, was absent from the adversarial literature, opening the door to new research direction either by proposing new initialization schemes to improve robustness while guaranteeing good generalization or new gradient-based weight updates to enforce the robustness of the model or yet again by tracking robustness metrics alongside the loss function throughout training.

## Acknowledgements

This work was partially supported by the Wallenberg AI, Autonomous Systems and Software Program (WASP) funded by the Knut and Alice Wallenberg Foundation. The computation (on GPUs) was enabled by resources provided by the National Academic Infrastructure for Supercomputing in Sweden (NAISS) at Alvis partially funded by the Swedish Research Council through grant agreement no. "2024/22-309". We furthermore want to thank Dr. Yassir Jedra for revising the manuscript and for a very helpful discussion on its different elements.

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

# Supplementary Material: If You Want to Be Robust, Be Wary of Initialization

## A  Proof of Theorem 2

**Theorem.** *Let $f : (\mathcal{A}, \mathcal{X}) \to \mathcal{Y}$ denote a graph-based function composed of $T$ GCN layers, where the initial weight matrix of the $i$-th layer is denoted by $W_0^{(i)}$. For adversarial attacks only targeting node features of the input graph, with a budget $\epsilon$, we have (in respect to Definition 1):*

$$\gamma = \epsilon \prod_{i=1}^{T} \left( 2^t \left\| W_0^{(i)} \right\| + 2^{t+1} \left\| W_*^{(i)} \right\| \right) \left( \sum_{u \in \mathcal{V}} \hat{w}_u \right).$$

*with $t$ being the number of training epochs and $\hat{w}_u$ denoting the sum of normalized walks of length $(T-1)$ starting from node $u$.*

*Proof.* Let's consider a graph-function $f$ that is based on $T$ GCN-layers. The gradient descent update at epoch $t$ for a layer $i$ is written as:

$$W_{t+1}^{(i)} = W_t^{(i)} - \eta \nabla \mathcal{L}(W_t^{(i)}).$$

Since we consider that our loss function $\mathcal{L}$ to be $L$-smooth, we have the following result:

$$\left\| \nabla \mathcal{L}(W_t^{(i)}) \right\| \leq L \left\| W_t^{(i)} - W_*^{(i)} \right\|.$$

Consequently, after $t$ training epochs, we can write:

$$\begin{aligned}
\left\| W_t^{(i)} \right\| &= \left\| W_{t-1}^{(i)} - \eta \nabla \mathcal{L}(W_{t-1}^{(i)}) \right\| \\
&\leq \left\| W_{t-1}^{(i)} \right\| + \eta L \left\| W_{t-1}^{(i)} - W_*^{(i)} \right\| \\
&\leq (1 + \eta L) \left\| W_{t-1}^{(i)} \right\| + \eta L \left\| W_*^{(i)} \right\|.
\end{aligned}$$

In addition, we have that $\eta \leq \frac{1}{L}$. Hence, by recursion, we find that:

$$\left\| W_t^{(i)} \right\| \leq (1 + \eta L)^t \left\| W_0^{(i)} \right\| + \sum_{h=0}^{t} 2^h \left\| W_*^{(i)} \right\| \tag{3}$$

$$\leq (1 + \eta L)^t \left\| W_0^{(i)} \right\| + 2^{t+1} \left\| W_*^{(i)} \right\|. \tag{4}$$

Giving that we are considering feature-based adversarial attacks, let $X$ denote the original node features and $X'$ denote the perturbed adversarial features. With an attack budget $\epsilon$, from the work [1], we have the following result:

$$\forall [A, X'] \in B\left([A, X], \epsilon\right), \|f(A, X) - f(A, X')\| \leq \prod_{i=1}^{T} \left\| W_t^{(i)} \right\| \epsilon \left( \sum_{u \in \mathcal{V}} \hat{w}_u \right). \tag{5}$$

with $\hat{w}_u$ denoting the sum of normalized walks of length $(T-1)$ starting from node $u$. Consequently:

$$\sup_{[A, X'] \in B([A, X], \epsilon)} \|f(A, X) - f(A, X')\| \leq \prod_{i=1}^{T} \left\| W_t^{(i)} \right\| \epsilon \left( \sum_{u \in \mathcal{V}} \hat{w}_u \right). \tag{6}$$

From Equations (3) and (6), we conclude that:

$$\sup_{[A,X']\in B([A,X],\epsilon)} \|f(A,X) - f(A,X')\| \leq \epsilon \prod_{i=1}^{T} \left[ 2^t \left\|W_0^{(i)}\right\| + 2^{t+1} \left\|W_*^{(i)}\right\| \right] \left( \sum_{u\in\mathcal{V}} \hat{w}_u \right).$$

We conclude that $f$ is $(\epsilon;\gamma)$-robust with:

$$\gamma = \epsilon \prod_{i=1}^{T} \left( 2^t \left\|W_0^{(i)}\right\| + 2^{t+1} \left\|W_*^{(i)}\right\| \right) \left( \sum_{u\in\mathcal{V}} \hat{w}_u \right).$$

$\square$

## B  Proof of Theorem 3

**Theorem.** *Let $f : (\mathcal{A}, \mathcal{X}) \to \mathcal{Y}$ denote a graph-based function composed of $T$ GCN layers, where the initial weight matrix of the $i$-th layer is denoted by $W_0^{(i)}$. Let $f$ be the number of used training epochs. When $f$ is subject to structural attacks, with a budget $\epsilon$, we have (in respect to Definition 1):*

$$\gamma = \epsilon \prod_{i=1}^{T} \left( 2^t \left\|W_0^{(i)}\right\| + 2^{t+1} \left\|W_*^{(i)}\right\| \right) \|X\| \left( 1 + T \prod_{i=1}^{T} \left( 2^t \left\|W_0^{(i)}\right\| + 2^{t+1} \left\|W_*^{(i)}\right\| \right) \right).$$

*Proof.* Similar to the previous proof, let's consider a graph-function $f$ that is based on $T$ GCN-layers and trained using gradient descent for $t$ epochs. We have the following result from Equation 3:

$$\left\|W_t^{(i)}\right\| \leq 2^t \left\|W_0^{(i)}\right\| + 2^{t+1} \left\|W_*^{(i)}\right\|. \tag{7}$$

For this proof, we are considering the model $f$ to be subject to structural perturbations. In this perspective, let $\tilde{A}$ denote the input non-attacked adjacency and $\tilde{A}'$ denote the attacked/perturbed adjacency, with $h'$ denoting its corresponding hidden representation. From the work [1], we have:

$$\forall [A', X] \in B([A,X], \epsilon), \|f(\tilde{A}, X) - f(\tilde{A}', X)\| \leq \prod_{i=1}^{T} \left\|W^{(i)}\right\| \|X\| \epsilon \left( 1 + T \prod_{i=1}^{T} \left\|W^{(i)}\right\| \right).$$

By combining the two previous results, we get the following inequality and hence the desired result:

$$\sup_{[A',X]\in B([A,X],\epsilon)} \|f(\tilde{A}, X) - f(\tilde{A}', X)\| \leq \epsilon \prod_{i=1}^{T} \left( 2^t \left\|W_0^{(i)}\right\| + 2^{t+1} \left\|W_*^{(i)}\right\| \right) \|X\|$$

$$\left( 1 + T \prod_{i=1}^{T} \left( 2^t \left\|W_0^{(i)}\right\| + 2^{t+1} \left\|W_*^{(i)}\right\| \right) \right).$$

$\square$

## C  Proof of Lemma 4

**Lemma.** *Let $f : (\mathcal{A}, \mathcal{X}) \to \mathcal{Y}$ denote a graph-based function composed of $T$ GCN layers for which the initial weight are drawn from the Gaussian distribution $\mathcal{N}(\mu, \Sigma)$. When subject to node features based adversarial attacks, we have the following:*

$$\mathbb{E}_{W_0 \sim \mathcal{N}(\mu,\Sigma)}[\mathcal{R}_\epsilon[f]] \leq \epsilon \prod_{i=1}^{T} \left( 2^t \sqrt{\mu^2 + tr(\Sigma)} + 2^{t+1} \left\|W_*^{(i)}\right\| \right) \left( \sum_{u\in\mathcal{V}} \hat{w}_u \right).$$

*Proof.* Let us consider $f$ to be a graph classifier based on $T$-GCN layers for which the initial weight are drawn from the Gaussian distribution. Specifically, $\forall i \leq L, W_0^{(i)} \sim \mathcal{N}(\mu, \Sigma)$. We have that:

$$\mathbb{E}\left[\left\|W_0^{(i)}\right\|\right] \leq \sqrt{\|\mu\|^2 + \text{tr}(\Sigma)}.$$

From Theorem 2, we have the following:

$$\gamma = \epsilon \prod_{i=1}^{T} \left(2^t \left\|W_0^{(i)}\right\| + 2^{t+1} \left\|W_*^{(i)}\right\|\right) \left(\sum_{u \in \mathcal{V}} \hat{w}_u\right).$$

Hence, combining the two elements results in the following:

$$\mathbb{E}_{W_0 \sim \mathcal{N}(\mu, \Sigma)}[\mathcal{R}_\epsilon[f]] \leq \epsilon \prod_{i=1}^{T} \left(2^t \sqrt{\mu^2 + \text{tr}(\Sigma)} + 2^{t+1} \left\|W_*^{(i)}\right\|\right) \left(\sum_{u \in \mathcal{V}} \hat{w}_u\right).$$

$\square$

## D   Proof of Theorem 5

**Theorem.** *Let $f : (\mathcal{A}, \mathcal{X}) \to \mathcal{Y}$ denote a graph-based function composed of $T$ GIN layers, where the initial weight matrix of the $i$-th layer is denoted by $W_0^{(i)}$. For adversarial attacks only targeting node features of the input graph, with a budget $\epsilon$, we have:*

$$\gamma = \prod_{l=1}^{T} \left(2^t \left\|W_0^{(i)}\right\| + 2^{t+1} \left\|W_*^{(i)}\right\|\right) \left[BT \max_{u \in \mathcal{V}} deg(u) + \epsilon\right].$$

*with $t$ being the number of training epochs and $deg(u)$ is the degree of node $u$.*

*Proof.* Let's consider a graph-function $f$ that is based on $T$ GIN-layers and trained using gradient descent for $t$ epochs. We have the following result from Equation 3:

$$\left\|W_t^{(i)}\right\| \leq (1 + \eta L)^t \left\|W_0^{(i)}\right\| + 2^{t+1} \left\|W_*^{(i)}\right\| \leq 2^t \left\|W_0^{(i)}\right\| + 2^{t+1} \left\|W_*^{(i)}\right\|. \tag{8}$$

Let $X$ denote the original node features and $X'$ the perturbed adversarial features. For an attack budget $\epsilon$, from the work [1], we have the following:

$$\forall [A', X] \in B([A, X], \epsilon), \|f(A, X) - f(A, X')\| \leq \prod_{l=1}^{T} \left\|W^{(l)}\right\| \left[BT \max_{u \in \mathcal{V}} deg(u) + \epsilon\right]. \tag{9}$$

Consequently, we can merge the two inequalities resulting in the following:

$$\gamma = \prod_{l=1}^{T} \left(2^t \left\|W_0^{(i)}\right\| + 2^{t+1} \left\|W_*^{(i)}\right\|\right) \left[BT \max_{u \in \mathcal{V}} deg(u) + \epsilon\right].$$

$\square$

# E    Proof of Theorem 6

**Theorem.** *Let $f : \mathcal{X} \subseteq \mathbf{R}^{in} \to \mathcal{Y} \subseteq \mathbf{R}^{out}$ be a $T$-layers neural network with $W_0^{(i)}$ denoting the initial weight matrix of the $i$-th layer. When subject to adversarial attacks, $f$ is $(\epsilon, \gamma) - robust$ with:*

$$\gamma = \epsilon \prod_{i=1}^{T} \left( 2^t \left\| W_0^{(i)} \right\| + 2^{t+1} \left\| W_*^{(i)} \right\| \right).$$

*Proof.* Let $f$ be a $T$-layers neural network. We additionally assume that its corresponding activation functions are 1-Lipschitz. Let $x$ (with $h$ its hidden representation) be an input vector and $x'$ (corresp. $h'$) its corresponding crafted adversarial input (corresp. hidden representation). For an adversarial attack with budget $\epsilon$, we have the following:

$$
\begin{aligned}
\forall x' \in \mathcal{X} : \|x - x'\| \le \epsilon, \|f(x) - f(x')\| &= \left\| h^{(l)} - h'^{(l)} \right\| \\
&= \left\| \phi^{(l)} \left( W^{(l)} h^{(l-1)} + b^{(l)} \right) - \phi^{(l)} \left( W^{(l)} h'^{(l-1)} + b^{(l)} \right) \right\| \\
&\le \left\| W^{(l)} \right\| \left\| h^{(l-1)} - h'^{(l-1)} \right\|.
\end{aligned}
$$

Recurrently, we find the final result as:

$$\sup_{x' \in \mathcal{X} : \|x - x'\| \le \epsilon} \|f(x) - f(x')\| \le \prod_{l=1}^{T} \left\| W^{(l)} \right\| \epsilon. \tag{10}$$

Note that similar results and analysis have been provided in previous work [6, 3]. By using the result derived in Equation 3, we have:

$$\left\| W_t^{(i)} \right\| \le 2^t \left\| W_0^{(i)} \right\| + 2^{t+1} \left\| W_*^{(i)} \right\|. \tag{11}$$

By merging these two inequalities, and applying the Markov Inequality, we find the following upper-bound:

$$\gamma = \epsilon \prod_{i=1}^{T} \left( 2^t \left\| W_0^{(i)} \right\| + 2^{t+1} \left\| W_*^{(i)} \right\| \right).$$

$\square$

# F    On the Case of Strong-Convexity - Proof of Lemma 7

**Lemma.** *Let $f : \mathcal{X} \subseteq \mathbf{R}^{in} \to \mathcal{Y} \subseteq \mathbf{R}^{out}$ be a $T$-layers neural network trained with a $\mu$-strongly convex and $L$-smooth loss function. Let $W_0^{(i)}$ denote the initial weight matrix of the $i$-th layer. When subject to adversarial attacks, with a budget $\epsilon$, we have that $f$ is $(\epsilon, \gamma) - robust$ with:*

$$\gamma = \epsilon \prod_{i=1}^{T} \left( (1 - \mu/L)^t \left\| W_0^{(i)} \right\| + 2 \left\| W_*^{(i)} \right\| \right).$$

*Proof.* We consider $f$ to be a $T$-layers neural network (following the same propagation as equation the one presented in Section 5). From Section E, we have the following:

$$\|f(x) - f(x')\| \le \prod_{l=1}^{T} \left\| W^{(l)} \right\| \epsilon.$$

In addition to the previous assumption of $L$-smoothness of the loss function, we consider that its $\mu$-strongly convex. Hence, for the layer $(l)$, we have the following result:

$$\left\| W_t^{(l)} \right\| \leq (1 - \mu/L)^t \left\| W_0^{(l)} - W_*^{(l)} \right\| + \left\| W_*^{(l)} \right\| \tag{12}$$

$$\leq (1 - \mu/L)^t \left\| W_0^{(l)} \right\| + 2 \left\| W_*^{(l)} \right\|. \tag{13}$$

When subject to adversarial attacks, we can use the previous result from E, specifically from Equation (10):

$$\sup_{x' \in \mathcal{X}: \|x - x'\| \leq \epsilon} \| f(x) - f(x') \| \leq \prod_{l=1}^{T} \left\| W^{(l)} \right\| \epsilon. \tag{14}$$

Hence, by merging the two previous results, we deduce that:

$$\gamma = \epsilon \prod_{i=1}^{T} \left( (1 - \mu/L)^t \left\| W_0^{(i)} \right\| + 2 \left\| W_*^{(i)} \right\| \right). \tag{15}$$

$\square$

Figure 5: Effect of the variance on the model's robustness in the case of Gaussian Initialization when subject to DICE (a,b) and Random Attacks (c,d) for both Cora and CiteSeer.

## G  Additional Results

### G.1  Adversarial Robustness of Deep Neural Networks

We consider the general family of neural networks for which the computation during layer $l$, using an activation function $\phi^{(l)}$, can be written as :

$$h^{(l)} = \phi^{(l)}(W^{(l)} h^{(l-1)} + b^{(l)}).$$

with $W^{(l)} \in \mathbb{R}^{n_{l-1}, n_l}$ being the weight matrix and $b_l \in \mathbb{R}^{n_l}$ the bias of the $l^{\text{th}}$ layer.

In this perspective, let $f : \mathbb{R}^{n_0} \to \mathbb{R}$ be a neural network $n_0$ being the input dimension. The adversarial task in this case consists of finding a perturbed input $\tilde{x}$ for which the prediction differs from the original prediction $f(x)$. The perturbed input $\tilde{x}$ should hence adhere to the similarity constraints defined by a perturbation budget $\epsilon$. Let's consider the $\ell_2$ norm within both the input space $\mathbb{R}^{n_0}$ and the output space $\mathbb{R}$, we can hence define the set of valid adversarial perturbation as:

$$B(x; \epsilon) = \{\tilde{x} : \|x - \tilde{x}\| \leq \epsilon\}.$$

Similar to Section 3, we can introduce the adversarial risk of a DNN within the input's neighborhood defined by the budget $\epsilon$ as the following:

$$\mathcal{R}_\epsilon[f] = \mathop{\mathbb{E}}_{x \sim \mathcal{D}} \left[ \sup_{\tilde{x} \in B(x;\epsilon)} \|(f(\tilde{x}) - f(x))\| \right]. \tag{16}$$

From this adapted adversarial risk, we can introduce the notion of a DNN's adversarial robustness

**Definition 8.** (DNN - Adversarial Robustness). The neural network $f : \mathbb{R}^{n_0} \to \mathbb{R}$ is said to be $(\epsilon, \gamma) -$ robust if its adversarial risk is upper-bounded by $\gamma$, i. e., $\mathcal{R}_\epsilon[f] \leq \gamma$.

### G.2 Additional Adversarial Attacks

In addition to the previously reported Mettack and PGD adversarial attack, we consider two additional adversarial attacks. Notably, we first consider "DICE" which involves iteratively perturbing a graph's structure by adding or removing edges while ensuring connectivity, and then adjusting the perturbation based on the gradient of the graph neural network's loss function to generate an adversarial example. The process aims to find a minimal perturbation that misleads the network's predictions while keeping the perturbation size small. We additionally consider a "Random" attack which consists of randomly perturbing the adjacency matrix by dropping or adding edges. Figure 5 shows the adversarial accuracy results on the Cora and CiteSeer dataset when subject to DICE and Random attacks for different values of $\sigma$ of the Gaussian initialization. Similarly, Figure 6 shows the effect of scaling both a uniform initialization and an Orthogonal one as previously explained in Section 6.

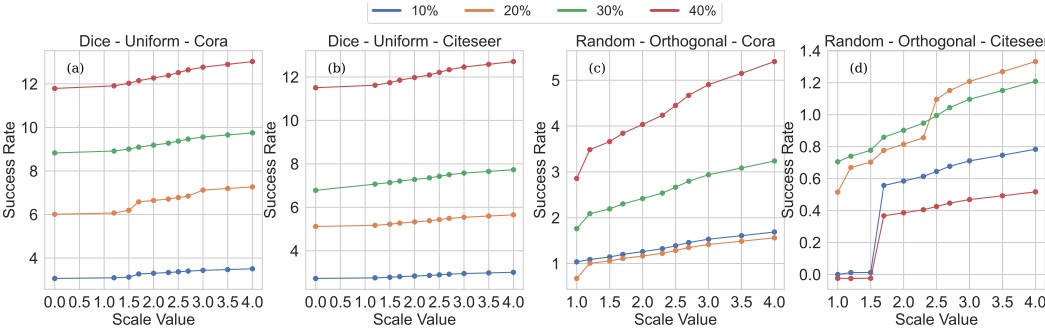

Figure 6: Effect of Uniform and Orthogonal Initialization on the model's robustness in the case of DICE Attack on Cora (a,c) and CiteSeer (b,d).

### G.3 Additional Datasets

We additionally extend the results to the ACM Dataset [31] within the node classification setting. Figure 7 presents the results using the Mettack, PGD and DICE for the ACM dataset for the Gaussian initialization (effect of $\sigma$), the Uniform and Orthogonal initialization.

### G.4 Additional Models

As previously explained in Section 5, while our theoretical analysis primarily focuses on GCN, GIN, and DNN models, the derived insights extend to other models as well. To illustrate this point, we examine the effect of initialization distribution on the performance of defense methodologies. Specifically, we first consider RGCN [39], which employs Gaussian distributions in its hidden layers to mitigate the effects of adversarial attacks. We additionally consider GCN-Jaccard [32] which preprocesses the network by eliminating edges that connect nodes with jaccard similarity of features smaller than a certain level. We use various initialization schemes, similar to those in our previous experiments, and evaluate against the same adversarial attacks (PGD, Mettack, and DICE). Figure 8 (resp. Figure 9) presents the adversarial accuracy and defense performance of RGCN (resp. GCN-Jaccard) on the Cora, CiteSeer, and ACM datasets. Although the performance gap is not very

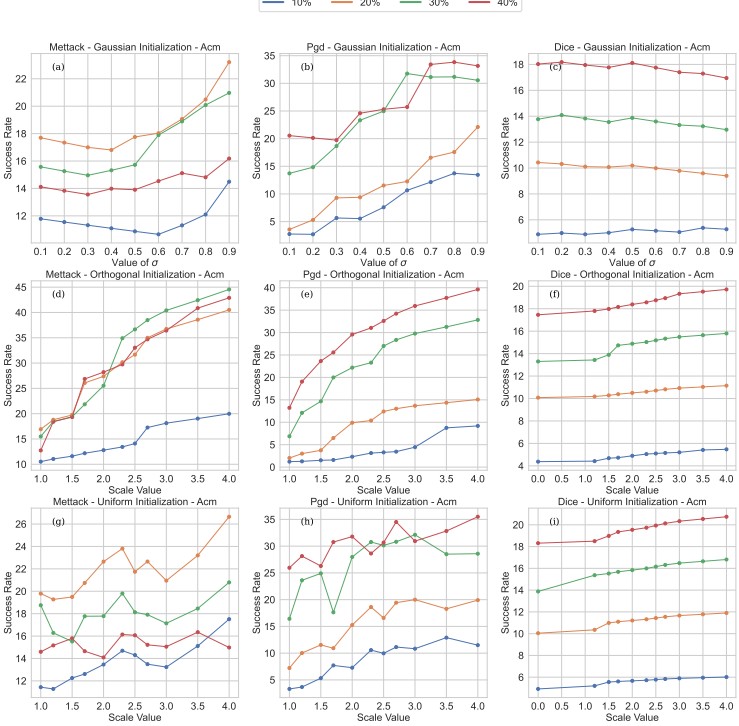

Figure 7: Effect of the Gaussian (a; b; c), Orthogonal (d; e; f) and Uniform (g;h;i) Initialization on the ACM dataset.

pronounced for Cora, it is clearly observed for CiteSeer and ACM. This demonstrates the broader applicability of our insights across different models but also defense methods.

## H   Datasets and Implementation details

**Datasets** Characteristics and information about the node classification datasets used in our experimental study are presented in Table 1. As outlined in the main paper, we conduct experiments on a set of citation networks, including Cora, CiteSeer (in the main paper), and ACM dataset (Appendix G) [31]. For all these datasets, we adhere to the train/valid/test splits provided by with the dataset.

**About the architectures.** In all of the experiments, the models employed a 2-layer convolutional architecture (consisting of two iterations of message passing and updating) stacked with a Multi-Layer Perception (MLP) as a readout. The intent was to compare the models in an iso-architectural setting, to ensure a fair evaluation of their robustness. We maintained the same hyperparameters, including a learning rate of 1e-2, 300 epochs, and a hidden feature dimension of 16 have been. To account for the impact of random initialization, each experiment was repeated 10 times.

**Reproducibility of the experiments.** We emphasize that all experiments should be easily reproducible by directly using the provided code. The archive contains a ReadMe file containing a small documentation on how to run the experiments.

Table 1: Statistics of the node classification datasets used in our experiments.

| DATASET | #FEATURES | #NODES | #EDGES | #CLASSES |
|---------|-----------|--------|--------|----------|
| CORA | 1433 | 2708 | 5208 | 7 |
| CITESEER | 3703 | 3327 | 4552 | 6 |

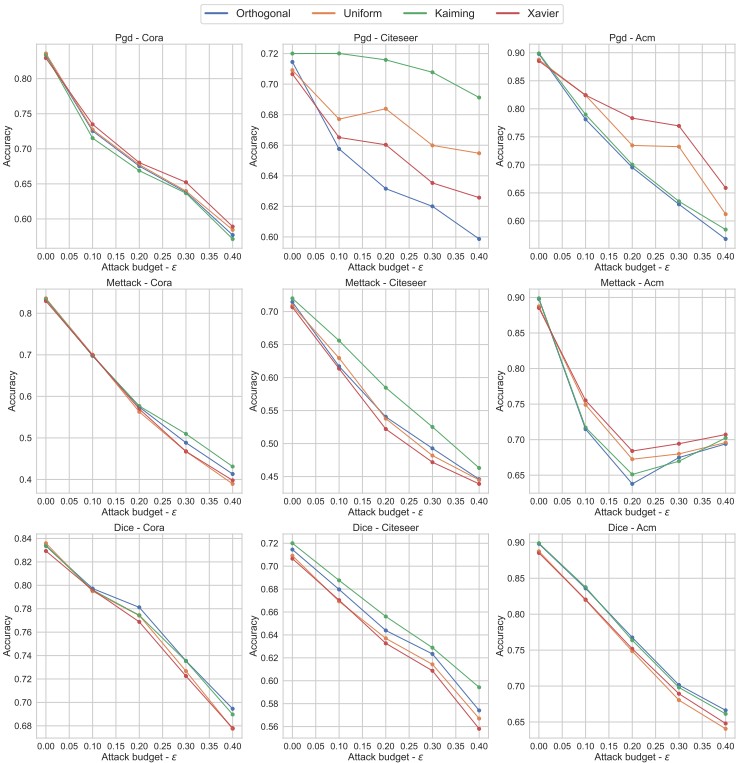

Figure 8: Effect of the initial distribution on RGCN's robustness and performance when subject to structural adversarial attacks.

**On the adversarial attacks.** For the PGD attack on the MNIST dataset, we used a step-size of $0.1$ and we set the number of iterations to $100$ (which was observed to be enough for the attack convergence). Note that we set these parameters for all the considered initializations in Figure 4 as our aim is to compare the effect of the different distribution on the final robustness.

**Implementation details.** Our implementation is available in the supplementary materials (and will be publicly available afterwards). It is built using the open-source library *PyTorch Geometric* (PyG) under the MIT license [12]. We used the publicly available implementation of the adversarial attacks provided in the DeepRobust package (https://github.com/DSE-MSU/DeepRobust). For RGCN, we used the implementation from the same package. The experiments have been run on both a NVIDIA A100 GPU where training a GCN takes around $1.2(\pm 0.2)$ s.

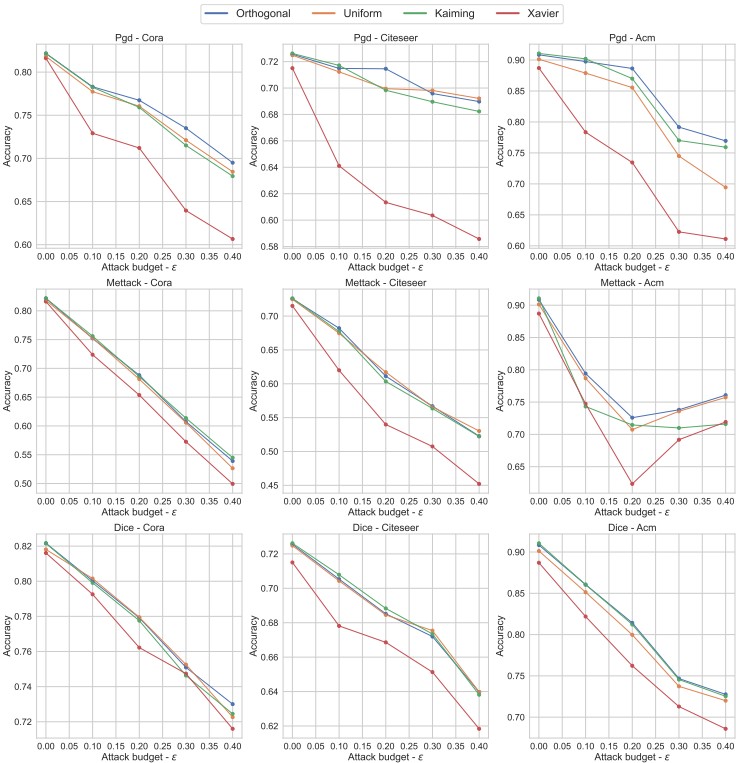

Figure 9: Effect of the initial distribution on GCN-Jaccard's robustness and performance when subject to structural adversarial attacks.

