# OpenReview forum: "If You Want to Be Robust, Be Wary of Initialization"
_NeurIPS.cc/2024/Conference — NeurIPS 2024 poster_

### Official Review · Reviewer_A9tB · 2024-07-11

**Soundness:** 3
**Presentation:** 3
**Contribution:** 2
**Rating:** 5
**Confidence:** 3

**Summary:**

This paper provides a theoretical study on the impact of number of epoch and initialisation to adversarial attack for GNN, potentially generalise to DNN. The theoretical evidence is supported by some empirical results.

**Strengths:**

The study of how number of training epoch and initialisation affect GNN robustness is novel and interesting.

The presentation is clear.

Both theoretical and empirical evidence are provided.

**Weaknesses:**

The study presents novel insights into Graph Neural Networks (GNNs), however, it is somewhat discussed in Deep Neural Networks (DNNs) regarding the number of training epochs [1, 2, 5] and initialisation methods [3, 4].

While the study is interesting, its contribution is somewhat limited due to the lack of proposed methods based on the findings.

From my understanding, the empirical validation is based on a single architecture, which may introduce limitations and bias. Given that the findings focus on initialisation, the empirical validation should cover a wider range of architectures.

The additional study on DNN and GIN is appreciated. However, the empirical evidence primarily focuses on outdated adversarial setups. The evidence would be more convincing if the authors provided empirical results using current and more practical adversarial setups, such as large-scale datasets (e.g., ImageNet-1K), modern architectures (e.g., ResNet, ConvNeXt, ViT), and recent adversarial attacks (e.g., AutoAttack, APGD) [5].

I believe that without the proposed method, validating the theoretical findings on a wider range of adversarial setups, as suggested, would help improve the contribution of this paper.

[1] Mo, Yichuan, et al. "When adversarial training meets vision transformers: Recipes from training to architecture." NeurIPS 2022.

[2] Pang, Tianyu, et al. "Bag of tricks for adversarial training." ICLR 2021.

[3] Hua, Andong, et al. "Initialization matters for adversarial transfer learning."  CVPR 2024.

[4] Vaishnavi, Pratik, Kevin Eykholt, and Amir Rahmati. "A Study of the Effects of Transfer Learning on Adversarial Robustness." Transactions on Machine Learning Research.

[5] Singh, Naman Deep, Francesco Croce, and Matthias Hein. "Revisiting adversarial training for imagenet: Architectures, training and generalization across threat models." NeurIPS 2023.

**Questions:**

Please see the weakness.

---

> ### Author Rebuttal · Authors · 2024-08-06
>
> We thank the reviewer for their constructive feedback and we would like to answer their main concerns and questions as follows:
>
> **[W1 - Regarding the provided references]** We thank the reviewer for these references that we weren’t aware of and we will include a complete analysis and discussion in our “Related work” section. It seems that the majority of these works are rather interested in the specific effect of initialization and other dynamics in the case of adversarial training. Our framework is rather more focused on the classical training and how different parameters (such as number of layers, number of epochs and learning rate) can affect the final underlying robustness. The majority of these proposed papers are also focusing on the empirical side hence our theoretical analysis can be of relevant to close the gap. In fact, we believe that our theoretical analysis can be adapted for the case of adversarial training (where we can consider that the “real” inputs have a gradient towards the downstream task and the “generated adversarial” inputs have a rather negative one).
>
> **[W2 - On the considered benchmarks]** We would like to thank the reviewer for seeing the value of the generalization section of our theoretical framework (which we note is not our main focus). We have drafted a more complete answer in our “General Rebuttal” as it was a common point between the reviewers. Typically, we wanted to underline that our main interest revolves around GNNs in which we have enough knowledge to make assumptions about the architectures and the internal dynamics of the Message-passing framework. For instance, in the specific case of GCN (Theorem 2), the upper-bound is dependent on the “normalized walks” within the input graph (meaning that denser graphs shall result in more adversarial effect). With a higher knowledge of the domain, similar theoretical results can be derived in the case of other domains such as NLP or Images resulting in more useful insights. Consequently, we believe that the introduced generalization part can be seen as pointers aimed to help close a gap which we have seen to be missing in the literature and which can be a first direction for other researchers to expand (such as the previously discussed adversarial training setting). Upon your suggestion (for which we are deeply grateful), we have initiated some new experiments to show the validity of our results on the ResNet family as well and to extend from MNIST (which was used for the DNN) to the Cifar-10 dataset. Table 1 reports the effect of the initialization on the ResNet model with both PGD and FGSM attacks. We will add these results (and expand them to other attacks) in our revised manuscript.
>
>
> Table 5: Effect of Initialization on ResNet when subject to FGSM and PGD attacks using CIFAR-10 Dataset.
> | Initialization | Clean Accuracy | FGSM($\epsilon=0.03$) | FGSM($\epsilon=0.07$) | PGD($\epsilon=0.03$) | PGD($\epsilon=0.07$) |
> |----------------|:--------------:|:---------------------:|:---------------------:|:--------------------:|:--------------------:|
> | Orthogonal     | 91.6 $\pm$ 0.2 |     44.1 $\pm$ 0.6    |     39.2 $\pm$ 0.5    |    21.9 $\pm$ 0.6    |    11.8 $\pm$ 0.5    |
> | Uniform        | 91.2 $\pm$ 0.3 |     46.8 $\pm$ 0.4    |     41.8 $\pm$ 0.3    |    24.3 $\pm$ 0.3    |    13.6 $\pm$ 0.4    |
> | Kaiming        | 92.3 $\pm$ 0.1 |     42.3 $\pm$ 0.2    |     36.9 $\pm$ 0.2    |    20.7 $\pm$ 0.4    |    10.1 $\pm$ 0.3    |
> | Xavier         | 92.1 $\pm$ 0.2 |     42.9 $\pm$ 0.3    |     37.6 $\pm$ 0.4    |    21.2 $\pm$ 0.5    |    10.6 $\pm$ 0.1    |
>
>
> We have also conducted a number of experiments on the interaction of our theoretical results with possible other defense methods (such as Adversarial training and TRADES [1]) which are in the same direction as the references you have provided.
>
> Table 6: Effect of Initialization on ResNet on the clean and attacked accuracy when subject to PGD adversarial training (AT-PGD) and TRADES with $\epsilon=8/255$.
> | Initialization | Clean Accuracy (AT-PGD) | Attacked Accuracy (AT-PGD) | Clean Accuracy (TRADES) | Attacked Accuracy (TRADES) |
> |----------------|:-----------------------:|:------------------------:|:-----------------------:|:------------------------:|
> | Orthogonal     |     83.8 $\pm$ 0.31     |      51.0 $\pm$ 0.37     |     82.6 $\pm$ 0.47     |      55.6 $\pm$ 0.32     |
> | Uniform        |     82.9 $\pm$ 0.17     |      54.1 $\pm$ 0.20     |     82.1 $\pm$ 0.23     |      57.9 $\pm$ 0.28     |
> | Kaiming        |     83.8 $\pm$ 0.22     |      46.5 $\pm$ 0.27     |     82.8 $\pm$ 0.27     |      52.4 $\pm$ 0.35     |
> | Xavier         |     83.2 $\pm$ 0.15     |      46.9 $\pm$ 0.19     |     82.7 $\pm$ 0.19     |      52.2 $\pm$ 0.23     |
>
>
> —
>
> [1] Zhang, Hongyang, Yaodong Yu, Jiantao Jiao, Eric Xing, Laurent El Ghaoui, and Michael Jordan. "Theoretically principled trade-off between robustness and accuracy." In International conference on machine learning, pp. 7472-7482. PMLR, 2019.

---

> > ### Comment · Reviewer_A9tB · 2024-08-08
> > **Official Comment by Reviewer A9tB**
> >
> > I thank authors for the rebuttal that address my major concerns. This is not further questions but rather a few suggestions that might improve the paper.
> >
> > - I understand the results for ImageNet-1K is not feasible for the rebuttal, but I strongly encourage the authors to include them in the future.
> >
> > - If the authors could compare the impact of initialisation on adversarial attacks with other defence mechanisms, it would underscore the significance of exploring initialisation for adversarial robustness. For example, varying the initialisations could result in a ~5% difference in attacked accuracy, while other defences might similarly reduce attacked accuracy by ~5%. This comparison would highlight the potential and importance of further investigating initialisation strategies for improving adversarial robustness in the future.
> >
> > Consequently, I increase my score to 5 borderline accept as my concern on contribution remains.

---

> > > ### Author Response · Authors · 2024-08-09
> > >
> > > We sincerely thank the reviewer for their thorough feedback and for taking the time to review our rebuttal. We agree that the comparison of the impact of different initialization schemes to the effectiveness of different existing defense methodology is an exciting future research direction. You provide an interesting perspective of our work, which we will gladly attempt to add to the revised manuscript.

---

### Official Review · Reviewer_ZfjT · 2024-07-12

**Soundness:** 3
**Presentation:** 4
**Contribution:** 3
**Rating:** 6
**Confidence:** 5

**Summary:**

This paper studies the impact of weight initialization (and training epochs) on adversarial robustness of a GNN model, both theoretically and empirically. The analysis is also extended to DNNs in general, although this is not the focus of this paper. The theoretical and empirical analysis both suggest that increase to norm of weights of initialization and increase in training epoch both have negative influence on adversarial robustness. The paper later compares different initialization strategies and show they lead to difference in robustness in their experiments.

**Strengths:**

This paper studies the influence of model initialization and training epochs to the adversarial robustness of the model, with both theoretical and empirical analysis. This paper also studies both GNN and DNN in general, with a great balance between staying focused, and comprehensiveness of the paper. The presentation of the paper is great and easy to follow.

**Weaknesses:**

Although there are many theories and claims presented in the first half of the paper, they can all simply be interpreted as "smaller norm of weights lead to more tightly bounded distance in output space given a same perturbation and thus better robustness", which is not that interesting. Especially, it seems that it did not consider the softmax function, or other type of normalization that may be applied.

Given these theories give very loose bound anyways, I am more interested to see how different initialization make a difference in practice. One big question of Fig.2 is that if they all trained to convergence and how their test accuracy look like. I hope to see more details about the experiment setting as well.

Section 6.4 is the most interesting and important section in my view, where it compares different initialization strategies and their influence on robustness. It is unfortunately too short and it could be made great if authors could expand the experiments, draw some conclusions, or reveal more deep insights on the choice of initialization. To prove that different initialization do lead to different robustness,  I think it need to cover more models and attacks. And importantly, as there is randomness in the initialization and training (if using dropout for example), I don't know if the results shown in Fig.4 is a single run of experiment or average among multiple runs. It should be average of multiple runs to avoid randomness which can lead to different conclusions.

I am happy to be convinced to the value of this paper if these major concerns are addressed.

**Questions:**

1. Regarding to Fig.2 , do they  all trained to convergence and how their test accuracy look like?
2. Regarding to Fig.4, is it a single run of experiment or average of multiple runs?

**Limitations:**

Presentation of limitations of this paper is decent.

---

> ### Author Rebuttal · Authors · 2024-08-06
>
> We thank the reviewer for their insightful feedback. We would like to answer their main concerns and questions as follows:
>
> **[W1 - General comment]** We feel slightly misunderstood by this comment and therefore want to apologize for any confusion that may have arisen from a possible lack of clarity in our explanations. In what follows, we try to address some of the perspectives that we feel weren’t fully clear:
>
> - Starting from our risk definition, we can fairly consider that if for an input point $x$, the output representation of the clean classifier and the attack classifier is close in the output space, then we can expect that the attack has failed (since the softmax function is just a monotone transformation based on the final output presentation). The motivation for not taking into account the softmax function in our setting was due to the desire to be general which is important in the case of GNNs since different downstream tasks are considered (node classification, node regression, graph classification/regression and link prediction).
> - “Smaller norm weights” was actually already investigated by other papers (such as Parseval regularization[1]). In our case, we rather consider how different training dynamics (such as number of epochs, the considered initialization and the learning rate) can affect reaching robustness. Hence, the novelty of our approach is focused on the training dynamics instead of making the statement that  “smaller weight norm” will help in robustness (which is already well investigated and well known in the adversarial literature).
> - Regarding the tightness of the bounds, the idea was to show the link between the previously discussed dynamics and the adversarial robustness. The provided bounds were actually the best we could come up without additional assumptions (making the study not relevant in practice). For instance, the bound can be tightened as stated after Theorem 2: " the dependence of $\gamma$ on $t$ can be sharpened by having $(1+\eta L)^t$ instead of $2^t$. With small $\eta$ (which is the case usually in practice), $(1+\eta L)^t \approx 1+ t\eta L$ resulting in a bound which depends linearly in $t$”.
>
> **[W2 - Regarding the generalization aspect]** We are grateful that this sub-result is of interest to you and the other reviewers. We have drafted a more detailed clarification to this point in our “General Rebuttal”. The main idea is that our interest revolves around GNNs in which we have enough knowledge to make assumptions about the architectures and the internal dynamics of the Message-passing framework. For instance, in the GCN’s case (Theorem 2), the upper-bound is dependent on the “normalized walks” within the input graph (meaning that denser graphs shall result in more adversarial effect). We believe that similar insights can be derived for other architectures such as those in the image domain. Consequently, the proposed generalization serves as a theoretical pointer to other researchers from the images or NLP domain to extend these results and eventually find more relevant and concise upper-bounds. We finally would like to refer the reviewer to the additional results on ResNet using the CIFAR 10 dataset in our general rebuttal (Table 1) and also the effect of adversarial training and other defenses such as TRACES on the same architecture (Table 2 - Response to Reviewer 1) which will be added to our revised manuscript.
>
> **[Q1 - On the convergence of the models]** All the considered models in Figure 2 were trained until convergence. Of course as pointed out, the initialization can have an effect of the clean accuracy. We have therefore chosen the distribution’s range of parameters (value of $\sigma$ of the gaussian distribution and the scale value in the Orthonormal and Uniform distributions) such as to be close to the state-of-the art performance of a GCN (with a maximum 2% margin). In the now following Table 4, we present the clean accuracy in the case of the extreme cases of the initial distributions:
>
> Table 4: GCN’s clean accuracy when subject to different initialization distribution using different parameters.
>
> |          | Gaussian ($\sigma=0.1$) | Gaussian ($\sigma=0.9$) | Uniform ($\text{scale}=1$) | Uniform ($\text{scale}=4$) | Orthogonal ($\text{scale}=1$) | Orthogonal ($\text{scale}=4$) |
> |----------|:-----------------------:|:-----------------------:|:--------------------------:|:--------------------------:|:-----------------------------:|:-----------------------------:|
> | Cora     |      83.5 $\pm$ 0.4     |      82.8 $\pm$ 0.9     |       83.8 $\pm$ 0.6       |       82.5 $\pm$ 0.8       |        84.1 $\pm$ 0.31        |         83.2 $\pm$ 0.5        |
> | CiteSeer |      73.1 $\pm$ 0.3     |      71.6 $\pm$ 0.7     |       72.8 $\pm$ 0.7       |       70.5 $\pm$ 0.5       |         72.4 $\pm$ 0.4        |         71.6 $\pm$ 0.6        |
>
> **[Q2 - On the number of trials]** We apologize if this wasn’t highlighted enough in our manuscript. As explained in our experimental setting (Section 6.1 and in details in Appendix H): “To mitigate the impact of randomness during training, each experiment was repeated 10 times, using the train/validation/test splits provided with the datasets.” We will add this also to the figure caption as well in the revised manuscript.
>
> —
>
> [1] Cisse, Moustapha, Piotr Bojanowski, Edouard Grave, Yann Dauphin, and Nicolas Usunier. "Parseval networks: Improving robustness to adversarial examples." In International conference on machine learning, pp. 854-863. PMLR, 2017.

---

> > ### Comment · Reviewer_ZfjT · 2024-08-08
> >
> > I appreciated the prompt response from the authors.
> >
> > [Q1] Thanks for the additional results. It looks like increase to the norm of initialization has negative impact of clean accuracy, which is as expected. So there is essentially a trade-off between accuracy and robustness (as most defensive method would have). I think this should highlighted in the paper. It would most interesting if we can get more general insights regarding to this trade-off, e.g. which does this trade-off compared to other defense methods; which initialization method achieves the best trade-off. Not say authors should address these now, but rather some extensive exploration directions to consider in the future.
> >
> > [Q2] Thanks for response. My question is addressed.

---

> > > ### Author Response · Authors · 2024-08-09
> > >
> > > We sincerely appreciate the reviewer’s thorough feedback and the time taken to review our rebuttal. We are pleased to have addressed their questions effectively. We will expand and add these insights to our revised manuscript.
> > > We also agree, that the mentioned areas are very interesting future research directions.

---

### Official Review · Reviewer_7mx1 · 2024-07-12

**Soundness:** 3
**Presentation:** 3
**Contribution:** 3
**Rating:** 7
**Confidence:** 4

**Summary:**

This work investigates the relationship between weight initialization and adversarial robustness, specifically in Graph Neural Networks (GNNs). In this setting, a defender wants  to train a GNN for which, given an input graph X, and adversary cannot find a similar graph which induces a different output from the GNN than X. The authors prove an upper bound on the adversarial risk of a given GNN that corresponds to the norm of the initial weights of the model. Bounds of this type are proved for both node feature robustness and structural robustness, with initialization shown to be more important in the structural case. Similar bounds are derived for other GNN architectures, like the GIN, and for DNNs more generally. Through experiments, it is shown that, as indicated by the theory, training models for more epochs can lead to higher robustness and lower variance parameters in weight initialization leads to higher robustness. These trends are shown to hold for different architectures, datasets, and weight initialization techniques.

**Strengths:**

This paper seemingly has high originality, being the first to study the impact of weight initialization on model robustness. It may have significant implications for the field of adversarial robustness, and may lead to improved adversarial training techniques. Clarity is one of this paper's main strengths. It is clear how the claims made in the introduction relate to the theoretical results, and the proofs provided in the appendix are easy to follow. Overall, I think this is a well-written paper with interesting results.

**Weaknesses:**

While the experimental section does validate many of the main results of the paper, there are other experiments that I would have liked to see to back up some claims made in earlier sections. Specifically, it would have been nice to see empirical results mirroring figure 2 but showing how different initializations impacted clean accuracy to show what the trade-off between accuracy and robustness looks like in a realistic setting.

Also, I think the framing and ordering of this paper may limit its reach. Based on the abstract, this paper seems to be targeted towards those who are interested in graph neural networks, despite the fact that the generalized results would be compelling even absent the discussion of GNNs. I feel this paper would be more effective if it put the generalized results (section 5) first, and then followed that up with a case study in GNNs to give an example of how this theory can be applied in specific settings.

Figures 1-3 are difficult to parse without closely reading their descriptions in the text. I would appreciate if the legends were made more descriptive and the captions were more detailed as to what is being shown.

**Questions:**

- How tight are the bounds presented in sections 4 and 5? Do you think future work might be able to tighten these bounds?
- In section G.1, it looks like the definition of adversarial risk is average-case (taking the expectation over all perturbations in a neighborhood), rather than worst-case. Was this intentional? It seems in conflict with the definition offered in equation 2.
- It seems like these results may also suggest that regularization (to target final weights with a smaller norm) is good for robustness. There have been recent papers that study the relationship between regularization and robustness [1,2], do you think your paper relates to this line of research?
- What are the implications of this for fine-tuning? Is this framework still useful when starting from a pretrained model?


[1] Nern, Laura F., et al. "On transfer of adversarial robustness from pretraining to downstream tasks." Advances in Neural Information Processing Systems 36 (2024).

[2] Dai, Sihui, Saeed Mahloujifar, and Prateek Mittal. "Formulating robustness against unforeseen attacks." Advances in Neural Information Processing Systems 35 (2022): 8647-8661.

**Limitations:**

Limitations are addressed in the problem setup and conclusion.

---

> ### Author Rebuttal · Authors · 2024-08-06
>
> We would like to sincerely thank the reviewer for their feedback and their comments.
>
> We are grateful for your comment on the novelty of our theoretical analysis and direction. Regarding the generalization of the results, we have drafted a complete response in the “General Rebuttal” as all the reviewers evoked this point. While this interest is certainly encouraging, we note that the framing of the paper was in direction with our main interest which revolves around GNNs in which we have enough knowledge to make assumptions about the architectures and the internal dynamics. For instance, in the GCN’s case (Theorem 2), the upper-bound is dependent on the “normalized walks” within the input graph (meaning that denser graphs shall result in more adversarial effect). We believe that similar insights can be derived for other architectures such as those in the image domain. Hence, this part of our manuscript can be seen as pointers to more advanced specific studies in each domain that can eventually result in interesting upper-bounds. Finally, we would like to refer the reviewer to the additional results on ResNet using the CIFAR 10 dataset in our general rebuttal (Table 1) and also the effect of adversarial training and other defenses such as TRACES on the same architecture (Table 2 - Response to Reviewer 1).
>
> In what follows, we try to address the additional questions/concerns raised by the reviewer:
>
> **[W1 - In respect to the clean accuracy of Figure 2]** In our experimental results, we have chosen to report the success of the attack since we thought it’s a better experimental representation of the theoretical results (our adversarial risk definition consists of the gap between the clean/attacked model). Nonetheless, ensuring a reasonable initial/clean accuracy (close to the state-of-the art by a 2% order) was important, hence the choice of the distribution’s range of parameters (value of $\sigma$ of the gaussian distribution and the scale value in the Orthonormal and Uniform distributions). Table 2 reports the clean accuracy in the case of the extreme cases of the initial distributions used in the analysis in Figure 2. We will add the complete figure of clean accuracies in the Appendix.
>
> Table 3: GCN’s clean accuracy when subject to different initialization distribution using different parameters.
>
> |          | Gaussian ($\sigma=0.1$) | Gaussian ($\sigma=0.9$) | Uniform ($\text{scale}=1$) | Uniform ($\text{scale}=4$) | Orthogonal ($\text{scale}=1$) | Orthogonal ($\text{scale}=4$) |
> |----------|:-----------------------:|:-----------------------:|:--------------------------:|:--------------------------:|:-----------------------------:|:-----------------------------:|
> | Cora     |      83.5 $\pm$ 0.4     |      82.8 $\pm$ 0.9     |       83.8 $\pm$ 0.6       |       82.5 $\pm$ 0.8       |        84.1 $\pm$ 0.31        |         83.2 $\pm$ 0.5        |
> | CiteSeer |      73.1 $\pm$ 0.3     |      71.6 $\pm$ 0.7     |       72.8 $\pm$ 0.7       |       70.5 $\pm$ 0.5       |         72.4 $\pm$ 0.4        |         71.6 $\pm$ 0.6        |
>
>
> **[Q1 - Regarding the tightness of the upper-bounds]** Indeed, these bounds can be tightened depending on the used optimization algorithm and the considered assumptions. In the case of Gradient Descent (GD) that we considered in the paper, we did not find better bounds for the iterates than those presented in the appendix (Eqs. (3) and (4) under non convex assumption and Eqs. (12) and (13) under strong convexity assumption). Regarding the final bounds we gave in sections 4, they can be tightened as stated after Theorem 2: " the dependence of $\gamma$ on $t$ can be sharpened by having $(1+\eta L)^t$ instead of $2^t$. With small $\eta$ (which is the case usually in practice), $(1+\eta L)^t \approx 1+ t\eta L$ resulting in a bound which depends linearly in $t$”.
>
> **[Q2 - On the definition of adversarial risk]** This is a small typo in writing, note that we have used the correct formulation in the Theorem’s proof (Appendix E - Eq. 10). We thank the reviewer for spotting this and we will correct it in the revised manuscript.
>
>
> **[Q3 - Linking to regularization techniques]** The idea of adversarial defense through regularization techniques is indeed very related to our work. The main difference is that in addition to the final weight’s norm, we also consider the initial weights, the learning rate and the number of epochs. Hence, our method can be seen as a generalization of studying the regularization techniques, in which we also focus on the training dynamics.
>
> **[Q4 - Extending to fine-tuning]** This is indeed a very interesting direction that could be seen as a sub-result of our theoretical analysis. For now, there isn’t much work on pre-trained GNNs, but we expect this field to expand in the coming years, which makes this question all the more relevant. In the case of other domains (such as NLP), there are two directions that could be investigated in this perspective. The first direction is when we consider that we have no control over the pre-trained model, hence the only control is over the classification/regression head that would be trained. In this case, our theoretical result is applied in the choice of the initial distribution of the weights and the number of fine-tuning epochs where a balance between accuracy and robustness should be found. In the second direction, where we have control over the pre-trained model, we don’t think our theoretical result can be applied since the pre-training consists of a number of “self-supervised tasks” (such as token masking or contrastive learning), and we believe that our adversarial risk definition doesn’t necessary extend from the pre-training embedding space to the downstream tasks. This latter point can be indeed an excellent research direction in which the goal would be to investigate the best pre-training dynamics capable of ensuring the downstream adversarial robustness of a pre-trained model.

---

> > ### Comment · Reviewer_7mx1 · 2024-08-08
> >
> > Thank you for your thorough response to my questions. I find the clean accuracies presented in Table 2 reassuring, and I appreciate the additional results validating the extension of the theory to different architectures. I will be raising my score to a 7.

---

> > > ### Author Response · Authors · 2024-08-09
> > >
> > > We are deeply grateful to the reviewer for taking the time to review our rebuttal. We are glad that we have been able to respond and address their concerns.

---

### Official Review · Reviewer_bCNj · 2024-07-13

**Soundness:** 3
**Presentation:** 3
**Contribution:** 3
**Rating:** 6
**Confidence:** 3

**Summary:**

The paper investigates the under-explored impact of weight initialization on the robustness of Graph Neural Networks (GNNs) against adversarial perturbations. The authors present a theoretical framework linking weight initialization strategies and training epochs to the model's resilience to adversarial attacks. They derive an upper bound that connects the model's robustness to initial weight norms and training epochs, showing that appropriate initialization strategies enhance robustness without degrading performance on clean datasets. The findings are validated through extensive experiments on various GNN models and real-world datasets subjected to different adversarial attacks, extending the theoretical insights to Deep Neural Networks (DNNs).

**Strengths:**

1. Originality:
    * The paper addresses a novel dimension of adversarial robustness in GNNs by focusing on weight initialization strategies, an area largely unexplored in existing literature.
    * Extends the theoretical framework to apply broadly to DNNs, showcasing its versatility and broader applicability.

2. Quality:

    * The theoretical analysis is rigorous, providing a clear connection between initialization strategies, training epochs, and model robustness.
    * Extensive experiments with diverse models and real-world datasets validate the theoretical findings, demonstrating the practical significance of the proposed framework.

3. Clarity:

    * The paper is well-structured, with a logical flow from theoretical analysis to experimental validation.

4. Significance:

    * The insights on the impact of weight initialization on adversarial robustness can influence future research directions and the development of more robust GNNs and DNNs.

**Weaknesses:**

1. Hyperparameter Tuning:
    * The paper does not discuss the sensitivity of the proposed initialization strategies to other hyperparameters, such as learning rate, batch size, or the number of layers. This omission could lead to challenges in replicating and generalizing the findings, as the effectiveness of the initialization strategies might vary with different hyperparameter settings.

2. Practical Implementation:
    * The practical implementation details of the proposed initialization strategies in real-world scenarios are not deeply explored. Practitioners might struggle to adopt these strategies without clear guidance on integrating them into existing workflows. Including a discussion on how to balance the trade-off between choosing the right number of epochs to achieve optimal clean accuracy and the most robust model would strengthen the paper.

**Questions:**

1. Interaction with Other Defense Mechanisms: how do the proposed initialization strategies interact with existing adversarial defense mechanisms, such as adversarial training or regularization techniques? For example, does combining your initialization methods with adversarial training lead to improved robustness, or are there any diminishing returns?

**Limitations:**

The theoretical framework is primarily validated on GNNs, leaving uncertainty about its applicability to other types of neural networks, such as recurrent neural networks (RNNs) or transformer models. This limitation reduces the perceived impact and versatility of the findings. Extending the theoretical analysis and experimental validation to include these other neural network architectures would enhance the generalizability of the results.

---

> ### Author Rebuttal · Authors · 2024-08-06
>
> We would like to sincerely thank the reviewer for their feedback and comments.
>
> As detailed in our “General Rebuttal”, the main focus of our research is related to GNNs, hence why the experimental setting was rather focused on this side. The extension to other models (such as DNNs in the paper or the additional ResNet on Cifar-10 that we provide in the general rebuttal - Table 1) was rather to show the extensibility of the theoretical results which we have seen to be missing from the corresponding literature. We will point out this further in our limitation section. In what follows we aim to address the raised concerns and questions one-by-one.
>
> **[W1 - Regarding the hyper-parameters]** Note that the provided theoretical results (Theorem 2 & 3) does indeed show the possible effect of the “number of layers” and “learning rate” (Equation 3 Section A of the Appendix). Hence, the reviewer is right on their intuition regarding the possible effect of hyper-parameters. Since the goal of our experimental setting (described in Appendix H) was aimed to validate the theoretical results, we followed the same hyperparameters that are “usually” used in the literature to ensure the classifier and the attacker’s convergence. Typically, we focused on a 2-layer GCN (since it can reach state-of the art accuracy and also because too many more layers result in the downgrade of the accuracy – which is known as over-smoothing in the GNN literature).
>
>
> **[W2 - Regarding the practical usage of the results]** This is indeed a fair and important point. Our theoretical analysis main result is that the dynamics of the training (Number of epochs, learning rate and the number of layers) also affect the final model’s adversarial robustness. Consequently, one possible future direction that could be investigated is a robustness metric to track the advancement of a model’s robustness (similar to the validation accuracy which is used to select the top model). Typically, one possible metric or measure would be to approach the proposed adversarial risk quantity (Equation2) using an estimator (such as randomised or stratified sampling from the input’s neighbourhood and evaluating the distance to the input’s prediction).
> To summarise, in addition to taking into account our proposed results (such as finding the right trade-off of epochs to reach convergence and adversarial robustness and choosing the right initialization), we need to investigate a model’s validation approach to keep track of its adversarial aspect. We will incorporate these practical guidelines  in our revised manuscript, serving to make our theoretical and experimental insights more useful in practice.
>
> **[Q1 - On the interaction with the adversarial defenses]** We have indeed studied the possible combination of our proposed insights with some “benchmark” graph adversarial defenses in Appendix G.4 where we studied RGCN (Figure 8) and GNN-Jaccard (Figure 9). For the sake of generalization, and we thank the reviewer for pointing out the idea, we have computed the results on adversarial training (based on PGD) and using TRADES [1] on the ResNet model using the Cifar-10 dataset. Table 2 reports the clean and attacked accuracy, showcasing therefore the application of our theoretical insights both to adversarial defenses but also to other architectures (besides GNNs and DNNs).
>
> Table 2: Effect of Initialization on ResNet on the clean and attacked accuracy when subject to PGD adversarial training (AT-PGD) and TRADES with $\epsilon=8/255$.
> | Initialization | Clean Accuracy (AT-PGD) | Attacked Accuracy (AT-PGD) | Clean Accuracy (TRADES) | Attacked Accuracy (TRADES) |
> |----------------|:-----------------------:|:------------------------:|:-----------------------:|:------------------------:|
> | Orthogonal     |     83.8 $\pm$ 0.31     |      51.0 $\pm$ 0.37     |     82.6 $\pm$ 0.47     |      55.6 $\pm$ 0.32     |
> | Uniform        |     82.9 $\pm$ 0.17     |      54.1 $\pm$ 0.20     |     82.1 $\pm$ 0.23     |      57.9 $\pm$ 0.28     |
> | Kaiming        |     83.8 $\pm$ 0.22     |      46.5 $\pm$ 0.27     |     82.8 $\pm$ 0.27     |      52.4 $\pm$ 0.35     |
> | Xavier         |     83.2 $\pm$ 0.15     |      46.9 $\pm$ 0.19     |     82.7 $\pm$ 0.19     |      52.2 $\pm$ 0.23     |
>
>
>
> —
>
> [1] Zhang, Hongyang, Yaodong Yu, Jiantao Jiao, Eric Xing, Laurent El Ghaoui, and Michael Jordan. "Theoretically principled trade-off between robustness and accuracy." In International conference on machine learning, pp. 7472-7482. PMLR, 2019.

---

> > ### Comment · Reviewer_bCNj · 2024-08-11
> >
> > I thank the authors for the detailed response. My concerns have been addressed. And I would like to remain my score.

---

> > > ### Author Response · Authors · 2024-08-12
> > >
> > > We are deeply grateful to the reviewer for taking the time to review our rebuttal. We are glad that we have been able to respond and address their concerns.

---

### Author Rebuttal · Authors · 2024-08-06

**General Comment to all the Reviewers:**

We are grateful to all the reviewers for their comments on the potential novelty of our theoretical insights and direction and we are also happy that our “generalization to other model” section has caught their attention. We would like to point out that our main research area revolves around GNNs (which in itself represents a vast research area from which different applications have emerged). In this perspective, our deep knowledge on the different architectures and adversarial attacks/defenses (which are different from the Image domain for instance, given the discrete aspect of the graphs) is within this area. The generalization part (Section 5) has been added as we have seen that our theoretical framework can easily be adapted to different architectures and that similar theoretical insights are missing from other domains (such as Images and NLP). In this direction, the adaptation to DNN (Theorem 3) aimed to give pointers and allow other researchers interested in these domains, which are more knowledgeable in architectures such as CNNs or RNNs, to unlock a new dimension in adversarial robustness.

Nonetheless,  upon specific request of some reviewers, we have provided additional results using ResNets in Table 1 on the Cifar-10 Dataset with both PGD and FGSM attacks in order to show the extension of the DNN’s previously provided results to this type of architecture. We will add the corresponding figures (exploring a range of $\epsilon$) to our manuscript.

We will additionally try to expand our results to the ImageNet dataset (which was a bit of struggle to manage for the rebuttal deadline given that each experiment is based on training 10 models of a specific initialization and evaluating the attacks).

Table 1: Effect of Initialization on ResNet when subject to FGSM and PGD attacks using Cifar-10 Dataset.
| Initialization | Clean Accuracy | FGSM($\epsilon=0.03$) | FGSM($\epsilon=0.07$) | PGD($\epsilon=0.03$) | PGD($\epsilon=0.07$) |
|----------------|:--------------:|:---------------------:|:---------------------:|:--------------------:|:--------------------:|
| Orthogonal     | 91.6 $\pm$ 0.2 |     44.1 $\pm$ 0.6    |     39.2 $\pm$ 0.5    |    21.9 $\pm$ 0.6    |    11.8 $\pm$ 0.5    |
| Uniform        | 91.2 $\pm$ 0.3 |     46.8 $\pm$ 0.4    |     41.8 $\pm$ 0.3    |    24.3 $\pm$ 0.3    |    13.6 $\pm$ 0.4    |
| Kaiming        | 92.3 $\pm$ 0.1 |     42.3 $\pm$ 0.2    |     36.9 $\pm$ 0.2    |    20.7 $\pm$ 0.4    |    10.1 $\pm$ 0.3    |
| Xavier         | 92.1 $\pm$ 0.2 |     42.9 $\pm$ 0.3    |     37.6 $\pm$ 0.4    |    21.2 $\pm$ 0.5    |    10.6 $\pm$ 0.1    |

---

### Decision · Program_Chairs · 2024-09-25

**Decision:**

Accept (poster)

**Comment:**

The paper explores the relationship of weight initialization and adversarial robustness. Many of the proposed insights extend beyond GNNs, which are the primary focus of this work. During the discussion period, the reviewers shared concerns over the experimental protocol and whether the attacks are on par with the more recent attacks used in the literature. I share those concerns and believe that the experimental protocol should be stronger. On the theoretical front, I encourage the authors to do a deeper search, since there are previous works that discuss the relationship between robustness and initialization, e.g., "Robustness in deep learning: The good (width), the bad (depth), and the ugly (initialization)" (NeurIPS'22).